# Capturing Recent *Mycobacterium tuberculosis* Infection by Tuberculin Skin Test vs. Interferon-Gamma Release Assay

**DOI:** 10.3390/tropicalmed9040081

**Published:** 2024-04-11

**Authors:** Jesús Gutierrez, Mary Nsereko, LaShaunda L. Malone, Harriet Mayanja-Kizza, Hussein Kisingo, W. Henry Boom, Charles M. Bark, Catherine M. Stein

**Affiliations:** 1Department of Population and Quantitative Health Science, Case Western Reserve University, Cleveland, OH 44106, USA; cmj7@case.edu; 2Uganda-CWRU Research Collaboration and Department of Medicine, School of Medicine, Makerere University, Kampala 7062, Uganda; mnsereko@mucwru.or.ug (M.N.); hmk@chs.mak.ac.ug (H.M.-K.); hkisingo@mucwru.or.ug (H.K.); 3Department of Medicine, Case Western Reserve University, Cleveland, OH 44106, USAwhb@cwru.edu (W.H.B.); 4Division of Infectious Diseases, MetroHealth Medical Center, Cleveland, OH 44109, USA; cmb148@case.edu

**Keywords:** tuberculosis, *Mtb* infection, tuberculin skin test, interferon-gamma release assay

## Abstract

Reductions in tuberculosis (TB) incidence require identification of individuals at high risk of developing active disease, such as those with recent *Mycobacterium tuberculosis* (*Mtb*) infection. Using a prospective household contact (HHC) study in Kampala, Uganda, we diagnosed new *Mtb* infection using both the tuberculin skin test (TST) and interferon-gamma release assay (IGRA). Our study aimed to determine if the TST adds additional value to the characterization of IGRA converters. We identified 13 HHCs who only converted the IGRA (QFT-only converters), 39 HHCs who only converted their TST (TST-only converters), and 24 HHCs who converted both tests (QFT/TST converters). Univariate analysis revealed that TST-only converters were older. Additionally, increased odds of TST-only conversion were associated with older age (*p* = 0.02) and crowdedness (*p* = 0.025). QFT/TST converters had higher QFT quantitative values at conversion than QFT-only converters and a bigger change in TST quantitative values at conversion than TST-only converters. Collectively, these data indicate that TST conversion alone likely overestimates *Mtb* infection. Its correlation to older age suggests an “environmental” boosting response due to prolonged exposure to environmental mycobacteria. This result also suggests that QFT/TST conversion may be associated with a more robust immune response, which should be considered when planning vaccine studies.

## 1. Introduction

Tuberculosis (TB) continues to be a major public health problem globally. In 2022, more than 10 million people became ill with TB and over 1.3 million succumbed to the disease. The WHO’s End TB Strategy aims to achieve a 75% reduction in TB mortality and a 50% reduction in the TB incidence rate by 2025 [1,2]. Although some regions in the world are on track to achieve these milestones, the African continent has fallen behind [1]. Faster reductions in TB incidence and deaths require improvements in multiple facets, including identification of those at risk of progressing to TB. Several studies have demonstrated that of those with latent *Mycobacterium tuberculosis* (*Mtb*) infection (LTBI) who progress to TB, most do so within the first two years post infection [3]. Furthermore, individuals with a recent *Mtb* exposure and conversion of their tuberculin skin test (TST) and/or interferon-gamma release assay (IGRA) have a higher risk for progression to TB [4,5,6]. Thus, identifying and treating individuals with recent *Mtb* infection, as opposed to those with prevalent LTBI, could be an important step in preventing TB.

Today, we rely on the TST or IGRA to diagnose *Mtb* infection. These tests have important limitations. First, the tests only infer the presence of *Mtb* based on a person’s T-cell response to *Mtb* antigens and are poor predictors of progression to TB [7]. Also, given that both tests use different stimuli and measure different immunologic responses, discordance rates are substantial [8]. Since T-cell responses persist, a positive result in either the TST or IGRA cannot distinguish between recent vs. remote *Mtb* infection [9,10,11,12,13,14], and some with a remote infection may have cleared the infection [3,6,9]. Lately, the IGRA has become the preferred test of choice to diagnose LTBI over the TST due to improved specificity and requiring only one visit to perform the test [15,16,17,18,19]. In a meta-analysis, the pooled sensitivity for the QFT reached 78% and the specificity among BCG-vaccinated and non-BCG-vaccinated persons was 96% and 99%, respectively [18]. This preference is demonstrated in the recommendations of many national TB programs in low-endemic countries [18,20] and has expanded to clinical TB research [21,22].

Despite the increased specificity for *Mtb* of the IGRA, the TST is still widely used in TB endemic areas for cost and simplicity reasons in the evaluation of latent *Mtb* infection (LTBI). Longitudinal studies [8], such as household contact (HHC) studies, with at least 6 months of follow-up and repeat testing can capture recent and/or new *Mtb* infection and offer a unique opportunity to evaluate both tests for sensitization to *Mtb* antigens [23]. In this study, we aimed to determine if the TST adds additional information and/or value to the characterization of IGRA converters. Given the discordance between the TST and IGRA, understanding the impact of this discordance in the identification of recent or new *Mtb* infection, i.e., TST/IGRA converters, is important. Second, optimal characterization of converters could help to maximize the capture of recent *Mtb* infection as well as the design of clinical studies to prevent new and treat recent *Mtb* infection [24]. Finally, if identification of recent converters is practical, focused preventive drug treatment could impact the incidence of TB.

## 2. Materials and Methods

### 2.1. Study Setting

The study is set in the greater Kampala metropolitan area, Uganda. The prevalence of active TB in Uganda is high, with recent estimates reaching 401 per 100,000 for individuals 15 years of age and older [25]. The prevalence in Kampala has been estimated to be even higher at 764 per 100,000 [26].

### 2.2. Study Population

Individuals suspected of having active pulmonary TB at participating health centers in Kampala were referred to the Uganda-Case Western Reserve University Research Collaboration clinic starting in 2015. The referred TB suspects underwent counseling and consent for a standard TB assessment under a separately approved protocol for ongoing TB studies within the Collaboration. Diagnosis of TB for the index case was established through a standard assessment, based on chest radiograph, history, physical examination, and sputum smear for acid-fast bacilli (AFB) or MTB/RIF GeneXpert^®^ (Cepheid, Sunnyvale, CA, USA) testing. If the assessment confirmed disease, these individuals with pulmonary TB (“index cases”) were asked for permission to approach their household contacts (HHCs) who were at least 15 years old to determine their interest in study participation. Index cases were not enrolled in this study. Upon receiving documented permission from the TB index case, HHCs were contacted to provide informed consent within 28 days (or 3 months of HIV+ HHCs) of the confirmed TB diagnosis of the index case. Consenting HHCs who met eligibility criteria (see below) were enrolled if the index case of their household was sputum smear-positive for AFB, and subsequently confirmed to have active TB by sputum culture or MTB/RIF GeneXpert^®^ positivity. TB treatment per national guidelines for index cases was started as soon as disease was suspected and did not wait for culture results. Recruitment for this study occurred from 29 June 2016 until 10 March 2020.

The study protocol was reviewed and approved by The Uganda National Council on Science and Technology and the institutional review board at the University Hospitals Cleveland Medical Center, with UHCMC IRB number: 01-16-21. Written consent was obtained from all participants aged 18 and older. Assent was obtained prior to screening from individuals aged 15 to 17 as well as written informed consent from their parent or guardian.

### 2.3. Inclusion Criteria for Household Contacts

In order to be enrolled in the study as an HHC, potential participants needed to meet the following criteria: 1. Informed consent or assent (where indicated) 2. Adults and minors aged 15 years and older; 3. Individuals living in the same building (house, hut, or apartment) or portion of the building as the index case, thereby sharing air space with the index case, for at least one week during the three-month period immediately preceding the diagnosis of TB in the index case; 4. Negative urine pregnancy test in women of child-bearing potential. Breastfeeding women were permitted to enroll in this study.

### 2.4. Exclusion Criteria for Household Contacts

In addition, potential participants were excluded from the study due to the following criteria: 1. TB or other febrile illness or uncontrolled disease; 2. Peripheral blood CD4 lymphocyte count < 200/mm^3^ for HIV+ participants; 3. Chest radiograph consistent with TB; 4. Expected to be unavailable for the 12-month follow-up period.

### 2.5. Study Design

This is an ongoing prospective longitudinal cohort study with a household contact design. HHCs aged 15 years and older who were living with the index cases and met the eligibility criteria as outlined above were enrolled. At baseline, each HHC underwent TB symptom screening, HIV testing, a pregnancy test for females, an IGRA test, and TST placement to be checked within 3 days (Appendix A). The IGRA test used was the QuantiFERON^®^-TB Gold (QFT, Qiagen, Hilden, Germany) or the QuantiFERON^®^-TB Gold Plus (QFT-Plus, Qiagen, Hilden, Germany), depending on availability during study follow-up. The QFT and QFT-Plus results were interpreted according to the manufacturer’s recommendations. The TST was performed by the Mantoux method (5 tuberculin units per 0.1 mL of purified protein derivative, Tubersol; Connaught Laboratories Limited, Willowdale, ON, Canada). TST positivity was defined by a maximum induration of 10 mm or greater, or an induration of 5 mm or greater in HIV+ individuals. In addition, each HHC underwent an individual risk assessment, a review of medical history, a complete physical examination, and a chest radiograph. Households were evaluated to determine if the house was a muzigo (the traditional muzigo in Uganda is a one-room dwelling in which all activities are performed apart from cooking) [27], whether the cooking took place inside or outside the house, number of inhabitants, number of rooms, and number of windows. All HHCs were followed for 12 months. Initially, follow-up visits occurred at 3, 6, and 12 months after enrollment. Subsequently, a 9-month visit was added to the protocol. During each follow-up visit, HHCs underwent a health check, a TB symptom screening, and a repeat IGRA for HHCs who had not become pregnant. A repeat TST was performed at 12 months for those HHCs who had a negative TST at enrollment. HHCs were offered preventive treatment if they had a positive TST or QFT result at baseline or after conversion of these tests during follow-up. Data collection and management for this paper were performed using OpenClinica open-source software, version 3.16 (OpenClinica LLC and collaborators, Waltham, MA, USA, www.OpenClinica.com).

### 2.6. Definitions

Once the HHCs completed follow-up, they were classified based on the following definitions (Figure 1):

A person with LTBI was defined as an HHC with all positive QFT and TST results.A “short-term resister” (“resister”) was defined as an HHC with all negative QFT and TST results throughout the course of the study. This definition was consistent with previous work in TB from different endemic settings [28,29,30,31,32].A QFT-only conversion was based on at least three QFT results, requiring a negative QFT at baseline followed by at least one positive QFT result during follow-up, accompanied by TST results that did not change throughout the study.A TST-only conversion was based on two TST results (baseline and month 12) and was defined as an original TST of < 10 mm (or < 5 mm for HIV+ individuals) followed by a TST reaction of ≥ 10 mm (or ≥ 5 mm for HIV+ individuals) on subsequent testing with an increase of ≥ 6 mm [33], accompanied by QFT results that remained positive or negative throughout the study.A QFT/TST conversion was defined as a conversion of both the QFT and TST as described above. There were six QFT converters (five QFT/TST converters and one QFT-only converter) who had only one positive QFT result because it occurred at month 12. These QFT converters were included in the analysis.

There were 30 HHCs with indeterminate or inconsistent QFT results who were classified by consensus into one of these categories based on the preponderance of the data, when possible (Appendix A). In the end, there were 21 HHCs whose QFT results remained uninterpretable, and these were excluded along with the 48 HHCs with incomplete follow-up. When comparing the HHCs included in the primary analysis versus those who were excluded as noted here, those who were included had a lower proportion of female participants (61.5% vs. 75.4%, *p* = 0.047) and a lower TB risk score (6 [6–7] vs. 7 [6–8], *p* = 0.02) (Appendix A).

### 2.7. Statistical Analyses

First, we examined the similarity between subjects classified using only the TST results or only the QFT results to illustrate the impact of discordance between the two tests. This analysis included all classified HHCs.

Second, we aimed to illustrate how the TST and QFT would define conversion differently among HHCs. Therefore, we did not include in this analysis those defined as concordant LTBI (both TST and QFT results were consistently positive) or discordant “resisters” (classified as “resister” by one of the tests but as LTBI by the other). This analysis only included the three types of converter groups (QFT-only converters, TST-only converters, and QFT/TST converters) and the “resister” group (HHCs who remained QFT negative and TST negative during the 12-month follow-up despite a high-level exposure to an infectious TB index case). “Resisters” were included in this analysis since they are known to have unique and robust immune responses [34], making them an appropriate control group. Comparisons were first made at the univariate level. Analysis focused on the following variables: epidemiological risk score (ERS), HIV status, body mass index (BMI), BCG status, quantitative IGRA values, quantitative TST values, type of housing (a marker of socioeconomic status), and clinical characteristics of the index case that have been associated with higher risk of transmission. The ERS consists of variables indicative of risk for *Mtb* exposure and infection and has been previously used in other studies examining the risk of LTBI [35,36]. Further details on the ERS can be found in the Appendix A. Comparisons were performed using the chi-square test, student’s *t*-test, analysis of variance, Mann–Whitney test, and the Kruskal–Wallis test. The chi-square test was used for categorical variable comparison, while other tests were used to examine continuous variables. Significance was assessed using a 0.05 alpha cut-off, using the Bonferroni correction to correct for multiple comparisons. When an omnibus comparison was significant, a pairwise comparison was performed to determine which pair differences contributed to this result. This univariate analysis was then repeated combining QFT-only converters and QFT/TST converters into a single category, QFT converters.

Third, a cluster analysis of all available variables using Gower distance was performed to determine if the subject clustering obtained was similar to the predetermined conversion groups. The purpose of this analysis was to examine whether clinical and epidemiological variables could better define subgroups of HHCs and if these groupings aligned with a TST and/or QFT-based definition. Additional details are found in the Appendix A.

Fourth, we evaluated the difference between TST-only conversion and QFT-conversion using a logistic regression model, where the dependent variable was a TST-only conversion event versus a QFT conversion event. The independent variables that were initially included were chosen based on the results of the univariate and cluster analyses described above. Once these were chosen, we proceeded to formulate our logistic regression models using two different methods. The first model contained all of the chosen independent variables. The second model was obtained using an automated backwards elimination process. The two models were compared using the Akaike information criterion (AIC) method and the one with the lowest AIC was chosen.

Finally, since previous studies have suggested different cut-off values for positivity (especially for the IGRA), we examined the quantitative changes in both the QFT and TST tests to better quantify changes associated with each of our conversion definitions. All analyses were performed using R (version 2023.03.1) [37].

## 3. Results

### 3.1. Differences in Classification among All Classified HHCs by Type of Test

This first comparison was performed on the 314 individual HHCs enrolled by January 2022 that could be defined as a converter, LTBI, or “resister” (Figure 1). We focused on how similarly subjects could be classified by the TST or QFT. Table 1 shows that if classification was based on the TST, there would be 63 converters. By the QFT, the number of converters would be 37. If both tests were used, there would be only 24 converters. The conversion rate for TST-only converters was 27.7% (39 out of 141 TST negatives at baseline), and for QFT-only converters, it was 9.7% (13 out of 134 QFT negatives at baseline). The conversion rate for QFT/TST converters was 22.2% (24 out of 108 TST and QFT negatives at baseline). The concordance rate for QFT and TST was lowest for converters (64.9%), compared to LTBI and “resisters” (82.8% and 74.2%, resp.). These analyses were restricted to subjects with consistent QFT and TST results according to our definitions.

### 3.2. Individual, Household, and TB Index Characteristics of TST-Only, QFT-Only, and TST/QFT Converters

The main analysis was performed on the three converter groups and “resisters” (Figure 1). Table 2 summarizes the characteristics and comparisons for each converter group with “resisters.” Converter groups did not differ in terms of sex, HIV positivity, BMI, presence of BCG scar, ERS, and smoking history. They did differ in age (*p* = 0.04). TST-only converters were older than “resisters”: median age 32 [20–47] vs. 23 [19.9–36.5], respectively, *p* = 0.006). This particular result was significant even after Bonferroni correction. TST-only converters also appeared to be older than the other two converter groups; however, this difference did not reach statistical significance.

In comparing household characteristics by conversion group (Table 2), a greater proportion of QFT-only converters (69.2%) lived in muzigos. A higher proportion of QFT-only converters (92.3%) cooked outside the home compared to the other groups. QFT-only converters lived in homes with fewer windows (1 window [0–2]) compared to QFT/TST converters (2.5 windows [1–6]) and “resisters” (3 windows [1–4]) (*p* = 0.003 and *p* = 0.003, respectively). This result was statistically significant after Bonferroni correction. There were no differences in the measure of crowdedness (people per room) and sleeping location (sleeping in the same room or the same bed).

Next, we examined the characteristics of the index case for the different converter groups (Table 3). Index cases of TST-only converters had more advanced lung disease (61.5%) than those of QFT-only converters (23.1%) (*p* = 0.04). Index cases of TST-only converters had a longer duration of cough than those of QFT/TST converters: 90 days [60–142.5] vs. 52.5 days [30–60] (*p* = 0.007). In contrast QFT/TST converters had a higher proportion of index cases with hemoptysis (33.3%) than TST-only converters (10.3%) (*p* = 0.04).

### 3.3. Identifying Predictors of QFT and TST Conversion

Next, we conducted a logistic regression analysis of QFT conversion vs. TST-only conversion to determine how they might differ epidemiologically (Table 4). Included variables were based on the univariate analyses in Table 2 and Table 3. We also included variables that were most influential in the cluster analysis using Gower distance (Appendix A). Based on backwards elimination, we arrived at a model containing six predictors, including age of the household contact, presence of cavitary lesions in the chest x-ray of the index case, extent of TB lung disease in the index case by chest x-ray, duration of index case cough in days, ERS, and number of people per room.

We found that the odds of undergoing TST-only conversion increased by 5% (95% CI [0.01–0.10]) for each additional year of age. The odds of TST-only conversion also increased by 55.9% (95% CI [0.08–1.4]) for each additional person per room. On the other hand, the odds of TST conversion decreased by 39% (95% CI [0.38–0.96]) for each point increase on the ERS. Based on an AUC of 0.81, this logistic regression model provided good discriminatory power (Table 4).

**Table 3 tropicalmed-09-00081-t003:** Univariate analysis comparing QFT-only, TST-only, and QFT/TST converters based on index case characteristics.

	QFT-Only Converters	TST-Only Converters	QFT/TST Converters	“Resisters”	*p*-Value (Test)	Relevant Pairwise *p*-Value
Sex (female)	8 (61.5%)	20 (51.3%)	17 (70.8%)	35 (48.6%)	0.26 (Fisher’s)	NA
BCG scar present	10 (76.9%)	30 (76.9%)	17 (73.9%)	36 (56.3%)	0.10 (Fisher’s)	NA
HIV positive	1 (10.0%)	4 (12.9%)	0 (0%)	12 (18.8%)	0.14 (Fisher’s)	NA
BMI	19 [17–21]	18.5 [17–21]	18 [17–20]	19 [18–20]	0.42 (KW test)	NA
Cavitary lesions present	11 (84.6%)	32 (82.1%)	22 (91.7%)	43 (60.6%)	0.006 * (Fisher’s)	TST-only vs. “Resister”: 0.03 * QFT/TST vs. “Resister”: 0.005 *
Advanced lung disease	3 (23.1%)	24 (61.5%)	14 (58.3%)	25 (35.2%)	0.01 * (Fisher’s)	TST-only vs. QFT-only: 0.04 * TST-only vs. “Resister”: 0.01 *
Positive Smear	8 (88.9%)	29 (100%)	20 (100%)	55 (100%)	0.08 (Fisher’s)	NA
Positive GeneXpert	13 (100%)	39 (100%)	24 (100%)	71 (98.6%)	1.00 (Fisher’s)	NA
Coughing	13 (100%)	39 (100%)	24 (100%)	71 (100%)	1.00 (Fisher’s)	NA
Cough duration (days)	60 [60–90]	90 [60–142.5]	52.5 [30–60]	60 [30–90]	0.03 * (KW test)	TST-only vs. QFT/TST: 0.007 * TST-only vs. “Resister”: 0.017 *
Fever	9 (69.2%)	30 (76.9%)	19 (79.2%)	54 (76.1%)	0.92 (*Χ*^2^)	NA
Fever duration (days)	42 [14–75]	60 [30–90]	30 [27.8–60]	30 [14–56.3]	0.10 (KW test)	NA
Productive sputum	13 (100%)	36 (92.3%)	24 (100%)	68 (95.8%)	0.55 (Fisher’s)	NA
Productive sputum duration (days)	30 [30–90]	60 [30–120]	30 [27.8–60]	30 [30–60]	0.11 (KW test)	NA
Purulent sputum	6 (46.2%)	22 (56.4%)	14 (58.3%)	37 (52.9%)	0.89 (*Χ*^2^)	NA
Purulent sputum duration (days)	45 [30–60]	52.5 [30–120]	30 [23.3–60]	30 [15–90]	0.25 (KW test)	NA
Hemoptysis	1 (7.7%)	4 (10.3%)	8 (33.3%)	7 (10.0%)	0.04 * (Fisher’s)	QFT/TST vs. “Resisters”: 0.02 * QFT/TST vs. TST-only: 0.04 *
Hemoptysis duration (days)	14 [14–14]	5 [2.5–12.8]	2 [1.5–5]	7 [2.5–10.5]	0.58 (KW test)	NA
Dyspnea	6 (46.2%)	20 (52.6%)	10 (41.7%)	38 (53.5%)	0.77 (Fisher’s)	NA
Dyspnea duration (days)	60 [37.5–60]	90 [30–160]	60 [37.5–90]	30 [30–82.5]	0.18 (KW test)	NA
Weight loss	12 (92.3%)	33 (84.6%)	18 (75%)	59 (83.1%)	0.62 (Fisher’s)	NA
Weight loss duration (days)	60 [27.8–78.8]	60 [30–90]	52.5 [30–71.3]	30 [30–82.5]	0.20 (KW test)	NA

Counts (percentages) or median [quartiles]. ERS: Epidemiologic risk score. KW: Kruskal–Wallis. *Χ*^2^: Chi-square test. * Statistically significant at *p* < 0.05. When all QFT converters were compared to TST-only converters and “resisters,” no new significant associations were identified (Appendix A).

**Table 4 tropicalmed-09-00081-t004:** Logistic regression model of predictors of TST-only conversion vs. QFT conversion.

Covariates	N	Odds Ratio	95% CI	*p*-Value
Age (years)	NA	1.05	1.01–1.10	0.020 *
Presence of cavitary lesions				
No	11	(Ref.)	(Ref.)	(Ref.)
Yes	65	0.29	0.055–1.310	0.118
Index cough duration (days)	NA	1.01	1.000–1.018	0.099
Epidemiologic risk score	NA	0.61	0.375–0.956	0.038 *
People per room	NA	1.559	1.085–2.404	0.025 *
Advanced lung disease				
Not far advanced	35	(Ref.)	(Ref.)	(Ref.)
Far advanced	41	2.58	0.818–8.866	0.115

* Statistically significant at *p* < 0.05. CI: Confidence interval.

### 3.4. Quantitative QFT and TST Values for TST, QFT, and TST/QFT Converters

Next, we compared QFT quantitative values at the time of QFT conversion (Figure 2). This showed that QFT-only converters had lower IFN-gamma secretion values at conversion than QFT/TST converters (2.03 IU [0.92–3.82] vs. 4.14 IU [1.73 vs. 10.0], *p* = 0.03). We also compared quantitative TST values (Figure 3). At baseline, QFT/TST converters had a median 0 mm baseline TST (range 0–0 mm), while TST-only converters had a higher median 2 mm baseline TST (range 0–8 mm) (*p* = 0.001). By month 12, these two groups were indistinguishable: median: 15.6 mm vs. 14.6 mm (*p* = 0.25).

## 4. Discussion

The goal of this study was to determine the advantages of using the TST in addition to the IGRA in assessing recent *Mtb* infection. Some of the results pointed to the TST overestimating conversion when compared to the QFT. On the other hand, when compared to QFT-only converters, QFT/TST converters appeared to represent a different immunological signature that would need to be explored further. As conversion becomes an endpoint of interest for clinical trials [24], these differences could be important for future study design.

When examining TST-only converters, we observed that they were more numerous than IGRA-only converters. This result was similar to what was seen in a 2014 HHC study from Brazil [38]. While this could be the result of BCG cross-reactivity, BCG cross-reactivity is minimal when vaccination occurs during infancy and the TST is performed at least 10 years afterward [39]. Given that all our HHCs were at least 15 years of age and that the median age of our TST-only converters was 32, BCG cross-reactivity was unlikely to be the only reason behind the higher TST conversion rate. TST-only converters also appeared to be older than the other converter groups. This association was present in the univariate analysis and confirmed by our logistic regression model. The positive association between TST positivity and age has been detected in previous studies across different populations and age groups [40,41,42,43,44,45]. All of these studies, however, were cross-sectional in nature and could not differentiate between recent TST conversion and long-standing *Mtb* infection. Therefore, our study is the first to identify an association between recent TST conversion, as opposed to prevalent LTBI, and increasing age that could be possibly linked to longer environmental exposure to non-tuberculosis mycobacteria (NTM) in the setting of a BCG-primed immune response to cross-reactive mycobacterial antigens.

Furthermore, the use of the TST in conjunction with the QFT allowed us to identify converters who appeared to have distinct immunologic reactions at the time of conversion. When we examined QFT/TST converters, we observed that the QFT/TST converters had a higher change in QFT quantitative values at the time of QFT conversion than QFT-only converters. This difference of values was maintained at month 12, though this result did not attain statistical significance. Similarly, the change in TST increment was larger in QFT/TST converters then in TST-only converters. Overall, these results were indicative of a more robust immunologic response in HHCs who convert both tests. This idea is supported by studies in Senegal and South Africa that found that a combination of both a positive TST and a positive IGRA appeared better at predicting TB than either response alone [46,47]. Indeed, the predictive value of TB for the TST and IGRA has been the focus of several studies and continues to be a controversial subject. Some studies have found no difference between these two tests with respect to predictive ability [47,48,49,50]. On the other hand, other studies have found that the IGRA is better at predicting subsequent TB [51,52]. In the end, it is unclear if the TST or the QFT on their own may offer an adequate predictive value for subsequent TB disease. However, the significance of the more robust immunologic response by combined QFT/TST converters is not clear and is deserving of further investigation or development of a better biomarker for recent and/or new *Mtb* infection.

Recently, the WHO approved the recommendation to use newer *Mtb* antigen-based skin tests (TBSTs), including the Cy-Tb (Serum Institute of India, Pune, India), the Diaskintest (Generium, Volginsky, Russian Federation), and the EC-Test or C-TST (Anhui Zhifei Longcom, Hefei, China) [53]. Like the TST, these tests use the Mantoux method of injecting antigen and reading for induration after 48–72 h. Unlike the TST, these tests use more specific *Mtb* antigens (ESAT-6 and CFP-10), which are also utilized by the QFT [53]. The TBSTs have been found to have similar sensitivity and specificity to the QFT, even when tested in high prevalence settings such as China and South Africa [54,55,56,57]. Consequently, we would expect that our QFT results would be comparable to TBST results if these had been used. However, further studies using TBSTs would be needed to confirm this hypothesis.

Our analysis suffered from some limitations. First, our small sample size limited the power of the study. Second, although we obtained IGRA results at different points during the follow-up, TST results were only obtained at the beginning and at the end of the follow-up. Third, as noted in the Methods section, the group of HHCs excluded from the analysis had a higher TB risk score than those who were included in the analysis. Although this may have affected the results of our study, it is difficult to ascertain the exact implications of this difference.

## 5. Conclusions

In conclusion, our findings indicate that TST converters are older. If this association is due to ongoing exposure to environmental mycobacteria, as we have speculated, then vaccine design studies that only use the TST to assess endpoints will likely overestimate *Mtb* infection. Second, our results indicate that other epidemiological variables do not distinguish TST from IGRA converters and, therefore, cannot help define “true” conversion. Third, combined QFT/TST converters appear to have a more robust immune response. Consequently, neither the TST nor IGRA alone may be sufficient to provide an adequate understanding of conversion/recent infection. Future studies using both of these tests will be needed to elucidate whether dual conversion is a useful endpoint for clinical and immunological studies.

## Figures and Tables

**Figure 1 tropicalmed-09-00081-f001:**
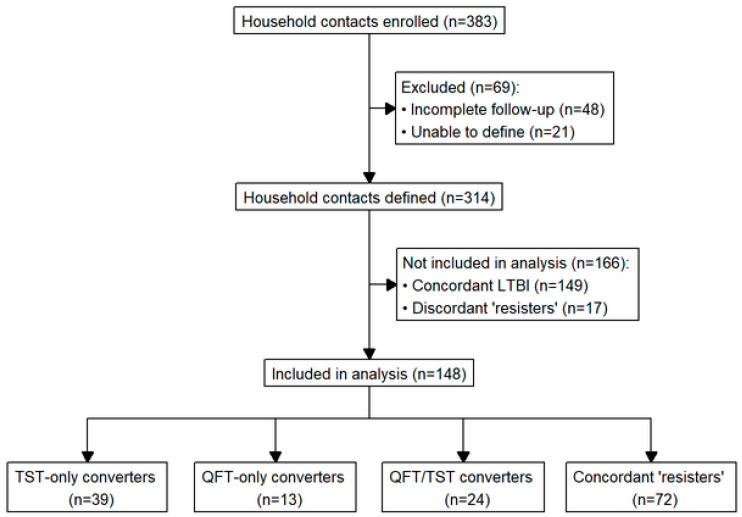
CONSORT diagram for this study. Out of 383 household contacts enrolled in our study, 69 were initially excluded because of incomplete follow-up or our inability to define them. A total of 314 household contacts were classified using both the TST and QFT. Those who were defined as concordant LTBI (both TST and QFT results were consistently positive) and discordant “resisters” (either two negative TST or all negative QFT) were excluded from analysis (n = 166). The remaining 148 household contacts were defined as follows: 39 TST-only converters (negative baseline TST with positive TST at 12 months), 13 were QFT-only converters (negative QFT at baseline followed by positive QFT during follow-up), 24 were QFT/TST converters (negative baseline TST and QFT followed by positive TST and QFT during follow-up), and 72 were concordant “short-term resisters” (all TST and QFT results remained negative). All QFT converters had a positive month 12 QFT result.

**Figure 2 tropicalmed-09-00081-f002:**
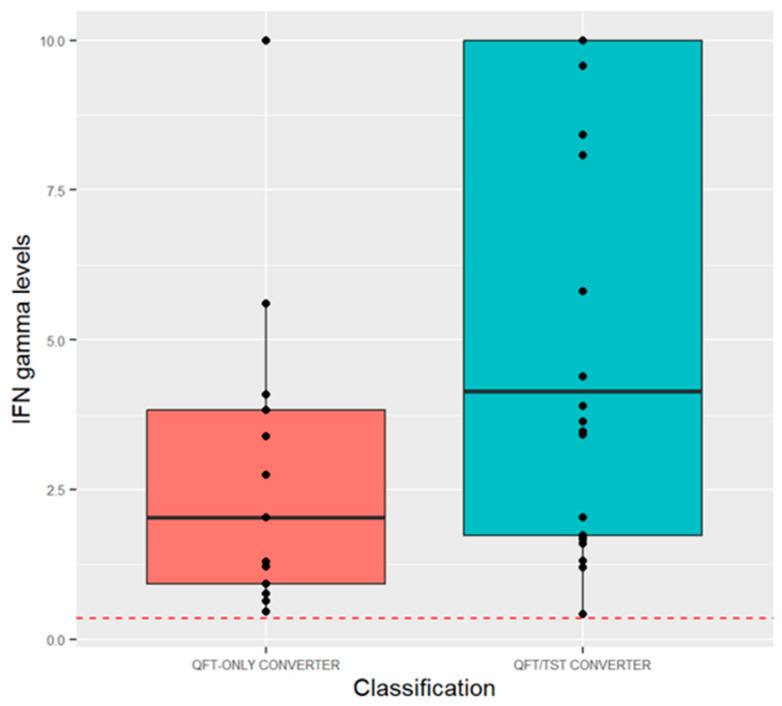
Quantitative QFT values of QFT/TST converters and QFT-only converters before and after QFT conversion. Red dotted line indicates 0.35 IU cut-off. If QFT plus was used, the highest value was used for this analysis. QFT-only converters had a lower value at conversion than QFT/TST converters (2.03 [0.92–3.82] vs. 4.14 [1.73–10.0], *p* = 0.03).

**Figure 3 tropicalmed-09-00081-f003:**
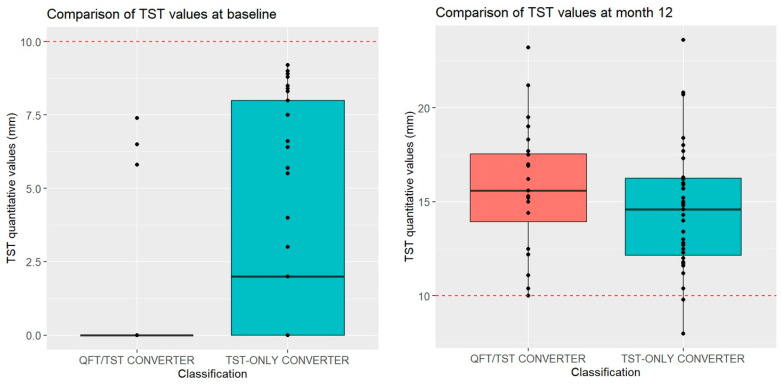
Quantitative TST values of QFT/TST converters and TST-only converters at baseline and at month 12 of follow-up. Red dotted line indicates 10 mm cut-off. At baseline, TST-only converters had a higher TST value than QFT/TST converters (2.0 [0.0–8.0] vs. 0.0 [0.0–0.0], *p* = 0.001). By month 12, this difference had disappeared (14.6 [12.2–16.3] vs. 15.6 [13.9–17.6], *p* = 0.25).

**Table 1 tropicalmed-09-00081-t001:** Cross-tabulation table of the discordance between QFT-only and TST-only classifications of converters, LTBI, and “resisters”.

	QFT-Based Classification				
TST-based Classification	N (%)	Converter	LTBI	“Resister”	Total
	Converter	24 (64.9%) ^a^	27 (15.0%) ^b^	12 (12.4%) ^b^	63 (20.1%)
	LTBI	11 (29.7%) ^c^	149 (82.8%) ^d^	13 (13.4%) ^e^	173 (55.1%)
	“Resister”	2 (5.4%) ^c^	4 (2.2%) ^e^	72 (74.2%) ^f^	78 (24.8%)
	Total	37 (100%)	180 (100%)	97 (100%)	314 (100%)

Of the 314 HHCs, 166 were excluded from the analysis, 39 were TST-only converters, 13 were QFT-only converters, and 24 were QFT/TST converters. In addition, 72 “resisters” were used as a control group. ^a^ These participants were classified as QFT/TST converters (n = 24). ^b^ These participants were classified as TST-only converters (n = 39). ^c^ These participants were classified as QFT-only converters (n = 13). ^d^ These participants were classified as concordant LTBI and were excluded from the analysis (n = 149). ^e^ These participants were classified as discordant “resisters” and were excluded from the analysis (n = 17). ^f^ These participants were classified as concordant “resisters” (n = 72).

**Table 2 tropicalmed-09-00081-t002:** Univariate analysis comparing QFT-only, TST-only, and QFT/TST converters based on individual and household characteristics.

	QFT-Only Converters	TST-Only Converters	QFT/TST Converters	“Resisters”	*p*-Value (Test)	Relevant Pairwise *p*-Value
N	13	39	24	72	NA	NA
Individual Characteristics						
Age	26 [20–36]	32 [20–47]	26.5 [23–39.3]	23 [19.8–36.5]	0.04 * (KW test)	TST-only vs. “Resister”: 0.006 *^¶^
Sex (female)	8 (61.5%)	21 (53.8%)	19 (79.2%)	43 (59.7%)	0.23 (Fisher’s)	NA
BCG scar present	12 (92.3%)	31 (81.6%)	14 (58.3%)	53 (75.7%)	0.08 (Fisher’s)	NA
HIV positive	0 (0%)	3 (7.7%)	3 (12.5%)	4 (5.6%)	0.57 (Fisher’s)	NA
BMI	22.2 [19.6–28.3]	23.6 [19.5–27.0]	22.5 [20.5–28.1]	22.8 [21.0–25.2]	0.93 (KW test)	NA
TB Risk score	7 [6–8]	6 [6–7]	7 [6–7]	6 [6–7]	0.15 (KW test)	NA
Quantitative IGRA values at conversion	2.0 [0.9–3.8]	3.4 [0.3–9.3]	4.1 [1.7–10]	NA	0.15 (KW test)	NA
No history of smoking	12 (92.3%)	37 (94.9%)	23 (95.8%)	61 (84.7%)	0.09 (Fisher’s)	NA
Spouse to Index	4 (30.8%)	7 (17.9%)	5 (20.8%)	8 (11.1%)	0.26 (*Χ*^2^)	NA
Household Characteristics						
Living in Muzigo	9 (69.2%)	20 (51.3%)	9 (37.5%)	21 (29.2%)	0.02 * (Fisher’s)	QFT-only vs. “Resister”: 0.01 * TST-only vs. “Resister”: 0.04 *
Cooking inside home	1 (7.7%)	10 (25.6%)	11(45.8%)	28 (38.9%)	0.05 * (Fisher’s)	QFT-only vs. QFT/TST: 0.03 * QFT-only vs. “Resister”: 0.03 *
Number of windows	1 [0–2]	1 [1–3]	2.5 [1–6]	3 [1–4]	0.007 *^#^ (KW test)	QFT-only vs. QFT/TST: 0.003 *^#^ QFT-only vs. “Resister”: 0.003 *^#^
People per room	2.5 [2–3]	2 [1.33–3]	1.67 [1–2.25]	2 [1.3–2.67]	0.08 (KW test)	NA
Sleeping in same room	11 (84.6%)	21 (53.8%)	15 (62.5%)	33 (45.8%)	0.053 (*Χ*^2^)	NA
Sleeping in same bed	5 (38.5%)	6 (15.4%)	6 (25%)	12 (16.7%)	0.23 (*Χ*^2^)	NA

Counts (percentages) or median [quartiles]. ERS: Epidemiologic risk score. KW: Kruskal–Wallis. *Χ*^2^: Chi-square test. * Statistically significant at *p* < 0.05. ^¶^ Statistically significant after Bonferroni correction at *p* < 0.006. ^#^ Statistically significant after Bonferroni correction at *p* < 0.008.

## Data Availability

The data presented in this study are available by applying to a data access committee chaired by Sudha Iyengar (ski@case.edu) and application to the Ugandan Institutional Review Board.

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
