# Peer review of "Capturing Recent Mycobacterium tuberculosis Infection by Tuberculin Skin Test vs. Interferon-Gamma Release Assay"

_tropicalmed, 2024, doi:10.3390/tropicalmed9040081_

Round 1

Reviewer 1 Report

Comments and Suggestions for Authors

We read with interest the manuscript entitled “Capturing Recent Mycobacterium tuberculosis Infection by Tuberculin Skin Test vs. Interferon-Gamma Release Assay”.

 Major comments:

 1. The authors aimed to determine if the TST adds additional value to the characterization of IGRA converters. The authors concluded that “neither the TST or IGRA alone may be sufficient to provide an adequate understanding of conversion/recent infection”.

Although the manuscript is structured, many information have been provided, and you need to be in the TB field to understand them and also try to see how they can be used in routine by TB programme. It would be good to simplify the whole manuscript (and mainly the methods) so that it can be read and understood easily by as many researchers / public implementers as possible?

2. Newer Mtb antigen-based skin tests (TBSTs) have been developed. TBSTs include the Cy-Tb (Serum Institute of India, India), Diaskintest (Generium, Russian Federation) and C-TST (formerly known as ESAT6-CFP10 test, Anhui Zhifei Longcom, China). All these tests use intradermal injection of antigen and, like the TST, are read after 48–72 hours as induration in millimetres, using the method suggested by Mantoux. In 2022 these tests have been recommended (WHO operational handbook on tuberculosis. Module 3: diagnosis - rapid diagnostics for tuberculosis detection).

 Although during the implementation of their study these tests were not recommended, it is important for the authors to include in their discussion a section in order to discuss their perspective by considering these new tests. They should also discuss how they think their results in this context could be relevant for TB Programme.

3. The follow-up of non-infected HHC has been performed during one year. Some HHC converted and the authors did not provide TPT. We think that it is not ethical although they mentioned that they received the approval from their ethical committee.

4. The characteristic of the 300 HHCs with indeterminate or inconsistent QFT results were not included. Their characteristic should be compared with the HHCs included in the study.

Minor comments.

 1. Line 93 (…. and subsequently confirmed to have active TB by sputum culture or MTB/RIF GeneXpert…):  The authors should mention that they do not wait sputum results to start TB Treatment.  

 2. Line 133-134 (Initially, follou-up ……after enrollment): When during the implementation of the study it has been decided to perform the 9 month visit. Did the authors inform the ethic committee? How many HHC received this 9-month visit?

 3. The table is not in the journal’s format.

 4. The format of some section are different from the journal’s recommendation. e.g. Table 4, (Not FAR ADVANCED), Line 300 (CI: Confidence Interval), Line 353 (at month 12 ….. to …….Similarly,),  

5. Line 313: The second bracket is missing for the p value.  

Reviewer 2 Report

Comments and Suggestions for Authors

It is a very good work, but I have some questions to the authors:

Given the response categories obtained in the study, what is the reason for the exclusion of the discordant “resisters” (either 2 negative TST or all negative QFT, n=17; rows 148 and 149)?

In the lines 177-180 you state: "...there were also 5 individuals with one positive QFT in the middle of the observation period who were excluded from the analysis...." Why not follow them too?

In the lines 257-259, you state: "They did differ in age (p=0.04). TST-only converters were older than “resisters”: median age 32 [20-47] vs. 23 [19.9-36.5], respectively, p = 0.006)".  It is very interesting that, even though the median age look different between these two groups, the 95%CI are very similar between them (practically the same). What do you think about this?

Round 2

Reviewer 1 Report

Comments and Suggestions for Authors

The authors have provided answers to all our concerns. The changes make the article easier to read.